



**How will the key marine calcifier *Emiliania huxleyi* respond to a warmer and more**
**thermally variable ocean?**
Xinwei Wang[1], Feixue Fu[2], Pingping Qu[2], Joshua D. Kling [2], Haibo Jiang[3], Yahui Gao[1,4],
David A. Hutchins[2*]
*1. School of Life Sciences, Xiamen University, Xiamen, 361005, China*
*2. Department of Biological Sciences, University of Southern California, Los Angeles,*
*California, 90089, USA*
*3. School of Life Sciences, Central China Normal University, Wuhan, Hubei, China.*
*4. Key Laboratory of the Ministry of Education for Coastal and Wetland Ecosystems,*
*Xiamen University, Xiamen 361005, China*
*Corresponding author: David Hutchins, tel: 1-213-7405616, fax: 1-213-7408123,
Email address: *dahutch@usc.edu*
Key words: thermal variation, *Emiliania huxleyi*, coccolithophore, calcification, growth
rate, elemental composition, global warming



## Abstract

Global warming will be combined with predicted increases in thermal variability in the future surface ocean, but how temperature dynamics will affect phytoplankton biology and biogeochemistry is largely unknown. Here, we examine the responses of the globally important marine coccolithophore *Emiliania huxleyi* to thermal variations at two frequencies (one-day and two-day) at low (18.5 °C) and high (25.5 °C) mean temperatures. Elevated temperature and thermal variation decreased growth, calcification and physiological rates, both individually and interactively. One-day thermal variation frequencies were less inhibitory than two-day variations under high temperature, indicating that high frequency thermal fluctuations may reduce heat-induced mortality and mitigate some impacts of extreme high temperature events. Cellular elemental composition and calcification was significantly affected by both thermal variation treatments relative to each other, and to the constant temperature controls. The negative effects of thermal variation on *E. huxleyi* growth rate and physiology are especially pronounced at high temperatures. These responses of the key marine calcifier *E. huxleyi* to warmer, more variable temperature regimes have potentially large implications for ocean productivity and marine biogeochemical cycles under a future changing climate.



## Introduction

Climate-driven changes such as ocean warming alter the productivity and
composition of marine phytoplankton communities, thereby influencing global
biogeochemical cycles (Boyd et al., 2018; Hutchins & Fu, 2017; Thomas, et al., 2012).
Increasing sea surface temperatures have been linked to global declines in
phytoplankton concentration (Boyce, et al., 2010), changes in spring bloom timing
(Friedland et al., 2018), and biogeographic shifts in harmful algal blooms (Gobler et al.,
2017). Warming and acidification may drive shifts away from dinoflagellate or diatom
dominance, and towards nanophytoplankton (Hare et al., 2007; Keys, et al., 2018).
Similarly, Morán et al. (2010) predicted that a gradual shift will occur towards smaller
primary producers in a warmer ocean.
Effects of temperature increases on phytoplankton diversity are uncertain.
Warming and phytoplankton biodiversity were found to be inversely correlated in a
coastal California diatom assemblage, at least on short timescales (Tatters et al., 2018).
In contrast, a five-year long mesocosm experiment found that elevated temperature can
modulate species coexistence, thus increasing phytoplankton species richness and
productivity (Yvon-Durocher et al. 2015). Globally, rising temperatures may result in
losses of phytoplankton biodiversity in the tropics, but gains in the polar regions
(Thomas et al., 2012). It is thought that ocean warming will lead to a poleward range
expansion of warm-water species at the expense of cold-water species (Boyd et al.,
2010; Gao et al., 2018; Hallegraeff, 2010; Hutchins & Fu, 2017; Thomas et al., 2012).
It is evident that rising ocean temperatures will benefit some groups, while having



detrimental consequences for others (Boyd et al., 2010, 2015, 2018; Feng, et al., 2017;
Fu et al., 2014). For example, recent decades of satellite observations show a striking
poleward shift in the distribution of blooms of the coccolithophore *Emiliania huxleyi*,
a species that was previously virtually absent in polar waters (Boyd et al., 2010;
Neukermans et al., 2018).

Coccolithophores are the most successful calcifying phytoplankton in the ocean,

and contribute almost half of global marine calcium carbonate production. They play
crucial biogeochemical roles by performing both photosynthesis and calcification, and
facilitate carbon export to the deep ocean through the ballasting effects of their calcium
carbonate shells (Klaas & Archer, 2002; Krumhardt et al., 2017; Monteiro et al., 2016).
*E. huxleyi* (Lohm.) is the most abundant and cosmopolitan coccolithophore, forming
prolific blooms in many regions (Holligan, et al., 1983; 1993; Iglesias-Rodríguez et al.,
2002; Westbroek et al., 1993).

The responses of *E. huxleyi* to global change factors have been intensively

investigated. Many *E. huxleyi* strains are sensitive to ocean acidification, which
negatively affects their growth rates and calcification (Feng et al., 2018; Hoppe et al.,
2011). However, among the many currently changing environmental drivers,
temperature may be among the most important in regulating coccolithophore
physiology (Boyd et al., 2010). Feng et al. (2008) reported that the growth rate of *E.*
*huxleyi* was improved by elevated temperature at low irradiance. Furthermore,
temperature was the most important driver controlling both cellular particulate organic
and inorganic carbon content of a Southern Hemisphere *E. huxleyi* strain (Feng et al.,



2018).

Most research about the effects of global warming on *E. huxleyi* and

phytoplankton in general has focused on predicted increases in mean temperatures.
However, in the natural environment, seawater temperatures fluctuate over timescales
ranging from hours, to days, to months (Bozinovic et al., 2011; Jiang et al., 2017).
Future climate models predict not only in an increase in mean temperature, but also an
increase in temperature variability (frequency and intensity), as well as a higher
probability of extreme events (IPCC 2014).

The impacts of climatic variability and extremes have been best studied in

metazoans, where they may sometimes have a larger effect than increases in climatic
averages alone (Vázquez et al., 2017; Vasseur et al., 2014; Zander et al., 2017).
Variability can promote greater zooplankton species richness, compared with long-term
average conditions (Cáceres 1997; Shurin et al. 2010). In corals, temperature variability
could buffer warming stress, elevate thermal tolerance and reduce the risk of bleaching
(Oliver & Palumbi, 2011; Safaie et al., 2018).

In comparison, we still lack a thorough understanding of how thermal variation

affects phytoplankton growth and physiology. Unlike zooplankton, the few available
studies suggest increasing thermal variation may decrease phytoplankton biomass and
biodiversity, and shift the community towards small phytoplankton (Burgmer &
Hillebrand, 2011; Rasconi et al., 2017). Two studies have shown that plastic responses
play a key role in acclimation and adaptation to thermal fluctuations in algae (Kremer
et al., 2018; Schaum & Collins, 2014). Population growth rates of phytoplankton in





fluctuating thermal environments have been quantitatively modeled based on data from
thermal response curves obtained under constant temperatures (Bernhardt et al., 2018).

In view of this relative lack of information on the effects of non-steady state

temperatures on biogeochemically important phytoplankton, we carried out a thermal
variability study using the Sargasso Sea *E. huxleyi* isolate CCMP371. Our experiments
combined ocean warming with thermal variations, with a focus on the increasing
frequency of temperature variations under global climate change. We examined growth
rates, photosynthesis, calcification and elemental composition under constant, one-day
and two-day temperature variations. This study is intended to provide insights into how
different frequencies of thermal variation may influence the physiology and
biogeochemistry of this important marine calcifying phytoplankton species under both
current and future sea surface temperatures.
**Materials and methods**

The marine coccolithophore *E. huxleyi* (Lohm.) Hay and Mohler stain CCMP371

(isolated from the Sargasso Sea) was maintained in the laboratory as stock batch
cultures in Aquil medium (100 µmol L$^{-1}$ NO$_3^-$ , 10 µmol L$^{-1}$ PO$_4^{3-}$) made with 0.2 µM-
filtered coastal seawater collected from the California region (Sunda et al., 2005).
Cells were grown at 22 $^{\circ}$C under 120 µmol photons m$^{-2}$ s$^{-1}$ cool white fluorescent light
with a 12 h/12 h light/dark cycle.

**Experimental set-up**

An aluminum thermal gradient block with a range of 13 temperatures was used to

perform the thermal response curve and temperature variation experiments. For the



thermal curve experiment, the extreme temperatures of the thermal-block were set to
8.5 °C and 28.6 °C, with intermediate temperatures of 10.5 °C, 12 °C, 13.5 °C, 15.5 °C,
17.5 °C, 18.5 °C, 21.3 °C, 22.6 °C, 24.5 °C, 26.6 °C, and 27.6 °C. The *E. huxleyi* cells
were transferred from the stock cultures into triplicate 120 ml acid washed
polycarbonate bottles in the thermal block under a 12 h light /12h dark cycle at 180
μmol photons m$^{-2}$ s$^{-1}$.
Semi-continuous culturing methods were used for all experiments. Cultures were
diluted every two days to keep them in exponential growth stage while acclimating to
the treatment temperatures for two weeks. Dilution volumes were calculated to match
growth rates of each individual replicate, as measured using in vivo chlorophyll a (Chl
*a*) fluorescence. Once steady-state growth rates were recorded for 3–5 consecutive
transfers, the cultures were sampled (Zhu et al., 2017). To estimate the negative growth
rates observed at 28.6 °C, these cultures were diluted from 22 °C stock cultures, and
sampled after 4-6 days for growth rates and elemental stoichiometry.
Six treatments were used to determine the responses of *E. huxleyi* growth,
photosynthesis and calcification to different frequencies of temperature fluctuation.
Temperature fluctuation treatments included:    1) Low temperature, constant (18.5
°C). 2) Low temperature, one-day fluctuation cycle (16-21°C, mean = 18.5°C). 3) Low
temperature, two-day fluctuation cycle (16-21°C, mean =18.5°C). 4) High temperature,
constant (25.5 °C). 5) High temperature, one-day fluctuation cycle (23-28°C, mean =
25.5°C). 6) High temperature, two-day fluctuation cycle (23-28°C, mean = 25.5°C).
The experimental *E. huxleyi* cultures were grown in triplicate in 120 ml acid washed



polycarbonate bottles using the thermal-block under a 12 h light /12h dark cycle at 180
μmol photons m$^{-2}$ s$^{-1}$.

For the variable temperature experiment, cultures were diluted semi-continuously

every two days for constant and one-day variation treatments, and every four days for
two-day variation treatments. 100 μmol L$^{-1}$ nitrate and 10 μmol L$^{-1}$ phosphate was
added every two days.    Cultures were grown for at least eight dilutions (~16 days for
constant and one day variation treatments; ~32 days for two-day variation treatments)
to acclimate to the different experimental conditions before final sampling.    All
variation treatments were sampled twice across the thermal variation cycle, once
during the cool phase and once during the warm phase.
**Growth rates**

In vivo fluorescence was measured daily for the one-day variation treatment and

every two days for the constant and two-day variation treatments using a Turner 10-
AU fluorometer (Turner Designs, CA). In vivo-derived growth rates were
subsequently verified using cell samples counted with a nanoplankton counting
chamber on an Olympus BX51 microscope. Specific growth rates (d$^{-1}$) were calculated
using the in vivo fluorescence and cell count data as: $\mu = \ln[N(T_2)/N(T_1)]/(T_2 - T_1)$, in
which $N(T_1)$ and $N(T_2)$ are the in vivo fluorescence values or cell counts at $T_1$ and $T_2$.
**Chl *a* analysis**

Twenty ml culture samples were filtered onto GF/F glass fiber filters (Whatman

GFC, Maidstone, UK) for Chl *a* analysis. In vitro Chl *a* was extracted with 90%
aqueous acetone for 24 hours at -20 °C, and then measured using a Turner 10-AU



fluorometer (Turner Design, USA). (Fu et al., 2007).
**Elemental analysis**
Elemental composition sampling included Total Particulate Carbon (TPC),
Particulate Organic Carbon (POC), Particulate Organic Nitrogen (PON), Particulate
Inorganic Carbon (PIC) and Particulate Organic Phosphorus (POP), allowing
calculation of cellular elemental stoichiometry and calcite/organic carbon rations
(PIC/POC) (Feng et al.; 2008). Culture samples for TPC, POC and PON, were
collected onto pre-combusted GF/F glass fiber filters (Whatman) and dried in a 60 ℃
oven overnight.   For POC analysis, filters were fumed for 24 hours with saturated
HCl to remove all inorganic carbon prior to analysis. TPC, PON and POC were then
measured by a 440 Elemental Analyzer (Costech Inc, CA) following Fu et al. (2007).
PIC was calculated as the difference between TPC and POC. For POP measurement,
culture samples were filtered on onto pre-combusted GF/F filters (Whatman) and
analyzed using a molybdate colorimetric method according to Fu et al. (2007).
**Total carbon fixation, photosynthetic and calcification rates & ratios**
Total carbon fixation, photosynthetic carbon fixation and calcification rates were
measured using $^{14}$C incubation techniques (Feng et al., 2008). Sixty mL culture
samples from each treatment were spiked with 0.2 μCi NaH$^{14}$CO$_3$ and then incubated
for 4 h under their respective experimental conditions. After incubation, samples were
filtered on two Whatman GF/F filters (30mL each) for total carbon fixation and
photosynthetic rate separately. The filters for photosynthetic rate measurement were
fumed with saturated HCl before adding scintillation fluid. Thirty mL from each



treatment (10 mL from each replicate bottle) was filtered immediately, after adding
equal amounts of NaH$^{14}$CO$_3$ for procedural filter blanks. Filters were then placed in 7
mL scintillation vials with 4 mL scintillation fluid overnight in the dark. To determine
the total radioactivity (TA), 0.2 μCi NaH$^{14}$CO$_3$ together with 100 μL phenylalanine
was placed in scintillation vials with the addition of 4 mL scintillation solution. All
samples were counted on a Perkin Elmer Liquid Scintillation Counter to measure the
radioactivity. Total carbon fixation and photosynthetic rate were calculated from TA,
final radioactivity and total dissolved inorganic carbon (DIC) values. Calcification rate
was then calculated as the difference between total carbon fixation and photosynthetic
rate for each sample.
**Model for population growth of *E. huxleyii***
Growth rates measured under constant temperatures in the thermal block were
fitted to the Eppley thermal performance curve or TPC (Eppley, 1972; Norberg, 2004;
Thomas et al., 2012). This function quantifies parameters of growth temperature
effects, including the temperature optimum for growth (T$_{opt}$), and high and low
temperature limits (T$_{max}$ and T$_{min}$ respectively) in our strain of *E. huxleyii*. A modified
version of this equation was also plotted to predict the impact that fluctuating
temperatures might have on growth rates at present-day and future mean temperatures
(Bernhardt et al., 2018, Kling et al. in review, Qu et al. in review).
**Statistical analysis**
The mean values of most parameters measured under the variation treatments were
calculated by averaging the values from the cool and warm phases, including all the



elemental content and ratios, photosynthetic and calcification rates and ratios. All
statistical analyses, including student t-tests and ANOVA were conducted using the
open source statistical software R version 3.5.0 (R Foundation).
**Results**
**Responses of *E. huxleyi* to warming**

The growth rates of *E. huxleyi* at constant temperature increased significantly with

warming from $0.09\pm0.01$ d$^{-1}$ at 8.5 °C to a maximum value of $0.90\pm0.02$ d$^{-1}$ at 21.3 °C.
Growth was optimal up to 24.5 °C, and then decreased rapidly to $-0.46\pm0.05$ d$^{-1}$ at 28.6
°C ($p<0.05$, Fig. 1).

The elemental ratios of the cells in the different temperature treatments were

compared to the average elemental ratios across the entire temperature range (Fig. 2).
The thermal trends of TPC/PON ratios were generally similar with those of growth
rates, in that ratios increased from 8.5 to 17.5 °C, and then decreased from 24.5 to 27.6
°C. The TPC/PON ratios at 8.5, 10.5 and 27.6 °C were significantly lower than the
average level of all the temperature points ($p<0.05$, Fig 2A). The POC/PON ratios of
most temperature points were very close to the mean value of 6.3, except at 27.6 °C
(7.1) and 28.6 °C (7.4), which were significantly higher than the average ($p<0.05$, Fig
2B). The highest PIC/POC ratio was $0.49\pm0.07$ at 22.6 °C, and the lowest PIC/POC
ratio was $0.05\pm0.04$ at 27.6 °C, a value that was almost 90% less than the highest value.
The PIC/POC ratios at the lowest temperature tested (10.5 °C) and at the high end of
the temperature range (26.6 and 27.6 °C) were significantly lower than the average
level (Fig. 2C). Chl *a*/POC ratios were significant lower at 8.5, 10.5 and 27.6 °C than



the mean, and at 17.5, 21.3, 22.6 and 24.5 °C were significantly higher than the average
(p<0.05, Fig. 2C). The trends of PIC/POC and Chl *a*/POC ratio were similar, in that
they gradually increased from low temperature and to the highest value at 22.6 °C, and
then dropped rapidly as temperature increased further. (Fig. 2C, D).
**Responses of *E. huxleyi* to temperature variations**

**Growth rate**

In low temperature experiments, both one-day and two-day temperature variations

had a negative effect on growth rate. The mean growth rates of the one-day
($0.71\pm0.01$ d$^{-1}$) and two-day ($0.72\pm0.01$ d$^{-1}$) variation treatments were not significantly
different from each other (p>0.05), but both were lower than that of the constant 18.5
°C treatment ($0.76\pm0.01$, $p < 0.05$) (Fig. 3A). Growth rates were low during the cool
phase (16 °C) of the experiment ($\sim$0.5-0.6 d$^{-1}$), but those of the two-day variation cycle
were not significantly different from the constant control at this temperature (p>0.05).
However, the cool phase of the one-day variation cycle had growth rates were lower
than those of the constant 16 °C treatment (p<0.05). During the warm phase of the
thermal cycle (21°C), there were no significant differences in the elevated growth rates
($\sim$0.85-0.9 d$^{-1}$) of the constant control and those of either variable treatment (p>0.05,
Fig. 3A).

In the high temperature experiments, as in the low temperature experiments, both

temperature variation frequencies had a negative effect on mean growth rates. The
growth rates in the two-day variation treatment were ($0.20\pm0.02$ d$^{-1}$), a decrease of
$\sim$74% compared with the constant 25.5 °C (p<0.05), and $\sim$62% of the one-day





variation treatment value (p<0.05, Fig. 3B). During the cool phase (23 $^{\circ}$C), the growth
rate of the one-day variation treatment was slightly lower (p<0.05) than the constant
23 $^{\circ}$C, but there were no significant changes between two-day variations and the
constant 23 $^{\circ}$C treatment (p > 0.05, Fig. 3B). During the warm phase (28 $^{\circ}$C), the
constant 28 $^{\circ}$C and two-day variation treatment both had negative growth rates of -
0.45±0.05 d$^{-1}$ and -0.45±0.04 d$^{-1}$, respectively.    However, the one-day variation
treatment had a low but positive warm phase growth rate at 0.25±0.02 d$^{-1}$ (Fig. 3B).
**Cellular PIC and POC contents and ratios**
In low temperature experiments, the cellular PIC content of the constant 18.5 $^{\circ}$C
treatment was 3.5±0.3 pg/cell, and there were no significant differences with
temperature variation treatments (p> 0.05, Table 1). However, the cellular POC
content of the constant 18.5 $^{\circ}$C treatment was 8.0±0.6 pg/cell, which was lower than
in the two-day variation treatment, but significantly higher than in the one-day
variation treatment (p<0.05).
Like POC, the PIC/POC ratio was significantly affected by temperature variations
(Fig. 4A). The lowest PIC/POC ratio was found in the one-day variation treatment
(0.38 ±0.07), which was significantly lower than the two-day variation treatment value
(p < 0.05), but close to that in the constant 18.5 $^{\circ}$C (p > 0.05). A similar trend was
found in both the cool (16 $^{\circ}$C) and warm phases (21 $^{\circ}$C) of the two variation treatments,
in that the PIC/POC ratio of the one-day variation treatment was lower than of the
two-day variation treatment (p < 0.05, Fig. 4A). Both variation treatments had lower
PIC/POC ratios during the warm phase than during the cool phase, although these





differences were not significant (p>0.05).

High temperature experiments showed particulate carbon trends that were contrary

to those of the low temperature treatments. The PIC content and PIC/POC ratios were
significantly decreased by temperature variation. The cellular PIC content of the
constant treatment (25.5 $^{\circ}$C) was 5.5±0.3 pg/cell, which was ~ 200% higher than that
of the one-day variation and ~ 160% higher than in the two-day variation treatments
(p<0.05, Table 1). The same trend was found for PIC/POC ratios in one-day variation
and two-day variation treatments, which decreased ~ 67% and 33% compared with the
constant 25.5 $^{\circ}$C treatment, respectively (p<0.05, Fig. 4B). However, the POC content
of one-day and two-day variation treatments was higher than in the constant 25.5 $^{\circ}$C
treatment (p < 0.05,Table 1). During the cool phase (23 $^{\circ}$C), the PIC content and
PIC/POC ratio of the one-day variation treatment was significantly lower than in the
two-day variation treatment, but contrary to PIC content, the POC content of the one-
day variation treatment was significantly higher than that in the two-day variation
treatment. During the warm phase (28 $^{\circ}$C), there were no significant differences of PIC
content, POC content, or PIC/POC ratio between the one-day and two-day variation
treatments (Fig. 4B, Table 1).
**Photosynthetic and calcification rates and ratios**

In low temperature treatments, there were no differences between total carbon

fixation rates (photosynthesis plus calcification) for the two variable treatments
relative to the constant control (Fig. 5A).   However, during the cool phase total
carbon fixation rates were higher in the one-day variation than in the two-day variation



(p<0.05, Fig 5A), while this rate was the same in both variation treatments during the
warm phase (p > 0.05, Fig. 5A). In high temperature experiments, the total carbon
fixation rates of the one-day and two-day variation treatments were significantly
decreased by about ~20% and ~18% respectively, compared with the constant 25.5 $^{\circ}$C
treatment (p<0.05, Fig. 5 B).
The photosynthetic and calcification rates of the constant 18.5 $^{\circ}$C treatment were
0.04±0.00 pmol C cell$^{-1}$ hr$^{-1}$ and 0.02±0.00 pmol C cell$^{-1}$ hr$^{-1}$, respectively, which were
not significantly different from both of the temperature variation treatments (p > 0.05,
Fig. 5 C,E). Photosynthetic rates changed within the thermal cycle for both one-day
and two-day variation treatments, with a decrease of 22% and 28% from the warm
phase to the cool phase, respectively (Fig. 5C). However, there were no significant
changes in calcification rates under either variation frequency treatment between the
cool and warm phases of the thermal cycles (p > 0.05).
In the mean 25.5 $^{\circ}$C experiment, photosynthetic rates were not significantly
different between the one-day variation and constant treatments (p > 0.05), while the
photosynthetic rate of the two-day variation was slightly higher than that of the
constant 25.5 $^{\circ}$C treatment (p<0.05, Fig. 5D). In contrast, calcification rates of one-
day and two-day variation treatments at a mean temperature of 25.5 $^{\circ}$ were
significantly decreased by about ~46% and ~51%, respectively, relative to the constant
control (p<0.05, Fig. 5F). There were no significant differences in total carbon fixation,
photosynthetic and calcification rates between the one-day variation and two-day
variation treatments during both the cool (23 $^{\circ}$C) and warm (28 $^{\circ}$C) phases (p>0.05,





Fig. 5 B,D,F).

In the low temperature treatments, there were no significant differences in

Cal/Photo ratios between the constant and the two variable treatments ($p > 0.05$, Fig
6A). In contrast, in the high temperature experiments, the Cal/Photo ratio of the one-
day variation and two-day variation treatments were decreased by ~40% and 49%,
respectively, compared with the constant 25.5 °C treatment ($p<0.05$, Fig. 6B). For both
low and high temperature experiments, there were no significant differences between
the one-day and two-day variation treatments in either the cool or warm phases of the
thermal cycle ($p > 0.05$, Fig. 6B). However, in both temperature treatments the lower
photosynthetic rates during the cool phase (Fig. 5C,D) resulted in an increase in the
Cal/Photo ratio during the cool phase for both the one-day and two-day variation
treatments ($p<0.05$ Fig. 6A,B).

**Elemental content and stoichiometry**

In the low temperature experiments, the one-day variation and two-day thermal

variations had different effects on cellular elemental contents and ratios, relative to the
constant 18.5 °C treatment. One-day variation increased most of the cellular elemental
and biochemical contents (TPC, PON, and Chl *a*) but with no significant difference
($p>0.05$), except for POP content ($p<0.05$), compared with the constant 18.5 °C
treatment (Table 1). In contrast, the two-day variation treatment decreased all the
measured cellular elemental and biochemical contents (TPC, PON, POP and Chl *a*,
$p<0.05$) in relation to the constant 18.5 °C treatment (Table 1). However, the



TPC/PON and Chl *a*/POC ratios of the two-day variation treatment were higher than
those of the one-day variation and constant 18.5 °C treatments (p<0.05, Fig. 7A,E),
while the PON/POP ratio was lower than in the one-day variation and constant 18.5
°C treatments (p<0.05, Fig. 7C). There were no significant differences in TPC/PON,
PON/POP and Chl *a*/POC ratios between the constant 18.5 °C and the one-day
variation treatments (p > 0.05, Fig. 7A).

In high temperature experiments, the highest cellular TPC, PON and POP contents

were all obtained under the one-day variation treatment, which was significantly
higher than under constant 25.5 °C conditions (p<0.05, Table 1). However, there were
no significant differences in cellular Chl *a* content between the constant 25.5 °C and
both variation treatments (p > 0.05, Table 1). The TPC/PON ratio of the constant 25.5
°C treatment was ~22% and ~35% higher than that of the two-day variation and one-
day variation treatments, respectively (p<0.05, Fig. 7B), while the PON/POP ratio was
highest in the day variation, followed by the two-day variation and finally by the
constant control (Fig. 7D). The Chl *a*/POC ratio of the one-day variation treatment
was significantly lower than that of the constant 25.5 °C and two-day variation
treatments (p<0.05), but there were no significant differences between the constant
25.5 °C and two-day variation treatments (p > 0.05, Fig. 7F).

During the cool phase of the high temperature experiments (23 °C), the cellular

TPC, PON, POP and Chl *a* content of two-day variation were all significantly lower
than in the one-day variation treatment (p<0.05). Similar decreasing trends during the
cool phase were observed for the TPC/PON ratios (Fig. 7B), but not the Chl *a*/POC



ratio, which was ~32% higher than in the one-day variation treatment ($p<0.05$, Fig.
7F). During the warm phase (28 $^o$C), there were no significant differences of cellular
TPC, PON and POP contents between one-day and two-day variation treatments ($p >$
0.05, Table 1) as well as the TPC/PON ratio (Fig 7B). However, the Chl *a* content of
the one-day variation treatment was ~20% lower than that of the two-day variation
treatment ($p<0.05$). The Chl *a*/POC ratio was not significantly different between the
one-day and two-day variation treatments at the warm phase ($p > 0.05$, Table 1, Fig.
7F).
**Experimental constant temperature performance curves and measured and**
**modeled fluctuating temperature TPCs**
The experimentally-determined constant condition TPCs and the predicted
fluctuating temperature condition TPCs based on the Eppley thermal performance
curve are shown in Fig. 8 for *E. huxleyi*. Compared with the measured TPC under
constant thermal conditions, the modeled TPC of the fluctuating temperature condition
showed a leftward shift towards lower temperatures at optimum temperatures and
above. The maximum and optimal temperature of the modeled fluctuating temperature
TPC were all lower than those of the measured constant condition TPC. In particular,
the optimal temperature for growth decreased from 22$^o$C in constant conditions to 21
$^o$C under fluctuating temperature conditions. At the same time, the maximum growth
rate ($\mu_{max}$) of the fluctuating temperature condition was 0.8 d$^{-1}$, which was lower than
the constant condition value of 0. 9 d$^{-1}$. The measured growth rates of experimental
one-day (0.71 d$^{-1}$) and two-day (0.72 d$^{-1}$) variation treatments at the relatively low



mean temperature of 18.5 °C closely matched the model-predicted fluctuating
temperature growth rate at this temperature (0.74$^{-1}$, Fig. 8). However, measured and
predicted growth rates did not match as well at the higher mean temperature. At 25.5
°C, the measured growth rate of the one-day variation was 0.52 d$^{-1}$, 30% higher than
the predicted fluctuating temperature growth rate of 0.40 d$^{-1}$. In contrast, the measured
growth rate of the experimental two-day variation treatment was 0.20 d$^{-1}$, a decrease
of 50% compared to the model-predicted fluctuating temperature growth rate of 0.40
d$^{-1}$ at this temperature (Fig. 8).
**Discussion**
**Effects of warming on *Emiliania huxleyi* growth rates and elemental ratios**
Thermal response curves and optimum growth temperatures describe the
importance of temperature as a control on the distribution of *E. huxleyi* strains in the
ocean (Buitenhuis et al., 2008; Paasche, 2001). The optimal temperature range of 21.3-
24.5 °C found in our study is similar to that of some other *E. huxleyi* strains (De Bodt
et al., 2010; Feng et al., 2017; Rosas-Navarro et al., 2016). Most studies have focused
on the lower part of the temperature curve where growth rates increase with rising
temperatures, with relatively few examining stressfully warm temperatures where
growth is inhibited (Feng et al., 2017; Matson et al., 2016). In our study, the descending
portion of the upper TPC ranged from 24.5 °C to 28.6 °C, at which point growth rates
became negative. This *E. huxleyi* strain was isolated from the Sargasso Sea where the
sea surface temperature can reach 29 °C in the summer, and will be higher in the future
with global warming (https://seatemperature.info/sargasso-sea-water-



temperature.html).    This suggests that this strain may be currently living near its upper
thermal limit for part of the year, as are many other tropical and subtropical
phytoplankton (Thomas et al. 2012), and that it may therefore be vulnerable to further
warming.

Calcification is the key biogeochemical functional trait of this species, and the

PIC/POC ratio of *E. huxleyi* can influenced by factors that include $CO_2$ concentration,
nutrient status, irradiance and temperature (Feng et al., 2008, 2017; Raven & Crawfurd,
2012). The cellular PIC/POC of *E. huxleyi* has been reported to decrease as irradiance
and $CO_2$ concentration rises, but increase under nitrate and phosphate limitation (Feng
et al., 2017; Paasche, 1999; Riegman et al., 2000). The effect of temperature on *E.*
*huxleyi* cellular PIC/POC ratio is however more complex. De Bodt et al. (2010) and
Gerecht et al. (2014) observed that higher cellular PIC/POC ratios were obtained at
lower temperatures for both *E. huxleyi* and *Coccolithus pelagicus*. Sett et al. (2014),
however, found an opposite trend, whereby the PIC/POC ratio increased with
temperature in another strain of *E. huxleyi*. Feng et al. (2017) reported that the cellular
PIC/POC of *E. huxleyi* was increased as the temperature rose from 4 °C to 11 °C, but
decreased with warming from 11 °C to 15 °C and remained steady afterwards.

In our study, the cellular PIC/POC ratio of *E. huxleyi* was positively correlated to

growth rate ($R^2$=0.73), and increased with warming from 8.5 °C to a maximum at 22.6
°C, and then decreased with further warming to 27.6 °C. In a meta-analysis of studies
using different coccolithophore subgroups, Krumhardt et al. (2017) found that the
highest PIC/POC ratios were observed between 15 °C and 20 °C, in the same thermal



range where the highest growth rates of *E. huxleyi* are found, as seen here and in Sett
et al. (2014). In contrast, Rosas-Navarro et al. (2016) reported that the cellular PIC/POC
ratio showed a minimum at optimal growth temperature (between 20 and 25 °C) for
three strains of *E. huxleyi*. However, the *E. huxleyi* strain used here was isolated from
a warmer area (the Sargasso Sea) compared with isolates from coastal Japan and New
Zealand in previous studies (Rosas-Navarro et al. 2016; Feng et al. 2017). The growth
temperature for our stock cultures was 22-24ºC, higher than that of the other two *E.*
*huxleyi* strains. Feng et al. (2017) also found that the optimal temperature for
calcification was close to the stock culture maintenance temperature in their study. Our
results also support suggestions that stressful high temperatures may lead to decreases
in cellular PIC/POC ratios and calcification (De Bodt et al., 2010; Feng et al., 2017;
Gerecht et al., 2014; Krumhardt et al., 2017).

The cellular Chl *a*/POC ratio of *E. huxleyi* showed a similar pattern with the

PIC/POC ratio, as it was also positively correlated to growth rate. Zhu et al. (2017)
reported the cellular Chl *a*/POC ratio of a Southern California diatom was also
correlated to growth rate across a very similar temperature range. In contrast, Feng et
al. (2017) found that the cellular Chl *a*/POC ratio of *E. huxleyi* dramatically decreased
with warming. However, in our experiments, the cellular Chl *a*/POC ratio was lower at
27.6 °C than at 28.6 °C, likely due to the negative growth rates and consequent lack of
acclimation of the cultures maintained at the highest temperature. Traits such as
PIC/POC ratios, Chl *a*/POC ratios and TPC/PON ratios also showed some evidence for
possible carryover from the stock cultures (22-24 °C) in this 28.6 °C treatment, as we





were forced to sample before the cells died completely, after only 2-3 cycles of dilution.
**Effect of thermal variation on *Emiliania huxleyi* growth and physiology**
*Constant vs variable temperature*

Thermal variability in the surface ocean is becoming an increasingly relevant

topic as global warming proceeds. In our study, we found that the growth rates of a
subtropical *E. huxleyi* strain were quite sensitive to temperature variation under both
low (18.5 °C, "winter") and high (25.5 °C, "summer") mean temperatures. In both low
and high temperature experiments, growth rates always decreased under temperature
variation, compared with the constant mean temperature. This result agrees with
previous studies showing that temperature variation slowed the growth rates of the
fresh water green alga *Chlorella pyrenoidosa* and the marine diatom *Cyclotella*
*meneghiniana*, as observed in laboratory work but also during long-term field
observations (Zhang et al., 2016).

This growth rate inhibition under temperature variation was more pronounced at

high temperature than at low temperature, indicating that variability at the warm range
boundary will have a stronger negative effect on population growth rate than
variability near the lower thermal limits (Bernhardt et al., 2018). This trend suggests
that high temperatures, whether constant or variable, can in general irreversibly
damage key cellular biochemical pathways and so inhibit growth rate. However,
following Jensen's inequality model to predict the thermal performance curve, there
should be an inflection point where the transfer between positive and negative effects
of temperature variability will occur compared with the constant thermal curve.



Conversely, for phytoplankton living in regions of suboptimal temperatures, thermal
variation can enhance growth (Bernhardt et al., 2018). Thus, for some polar
phytoplankton or for temperate species extending their ranges poleward, such as *E.*
*huxleyi* (Neukermans et al., 2018), not only warming but also thermal variability may
need to be taken into consideration in order to understand changes in high latitude
microbial communities and biogeochemistry cycles.

Temperature variation affected the physiology of *E. huxleyi* differently compared

with constant temperature. Physiological traits that were affected by thermal
fluctuations also differed at low temperature ("winter") and high temperature
("summer"), suggesting different response mechanisms. Under low temperature
variations (16-21 $^{\circ}$C), photosynthesis and calcification were correlated with
temperature, leading to rates similar to those observed with constant temperature.
However, elemental contents and ratios under thermal variations differed from
constant temperature. For instance, the cellular POC, PON, POP and Chl *a* contents
increased during one-day variations but decreased during two-day variations,
compared with constant temperature.

These cellular quota changes were reflected in elemental ratio differences

(PIC/POC, Chl a/POC and TPC/POC) between the thermal variation treatments and
constant temperature. However, the changes between thermal variation and constant
treatments were not significant under low temperature ("winter"), indicating that the
thermal variation wouldn't significantly influence biogeochemical cycles under these
conditions. Unlike constant temperature treatments where selection may favor a higher





growth rate, the trade-off for the thermal variation treatments may involve sacrificing
increased growth rate in order to adjust cellular stoichiometry to adapt to the
fluctuating environment.
In contrast, photosynthetic and calcification rates under high temperature thermal
variations (23-28 °C) were significantly different from those seen under constant
temperature (25 °C), especially the calcification rate. Thermal variation treatments
transiently but repeatedly experienced the extreme high temperature point (28 °C),
leading to extremely low calcification rates and PIC contents, and thus relatively low
PIC/POC and Cal/Photo ratios. Previous *E. huxleyi* studies agree that high temperature
decreases PIC content, PIC/POC ratios and Cal/Photo ratios (Feng et al., 2017; 2018;
Gerecht et al., 2014). The two different patterns of responses to thermal variation we
observed under low and high temperatures imply a seasonal pattern in the ways that
thermal variations will affect the elemental stoichiometry of *E. huxleyi* .
Under other stresses such as nutrient limitation, trade-offs between growth rates
and resource affinities may be necessary to adapt to thermal changes. For instance,
nitrate affinity declines in cultures of the large centric diatom *Coscinodiscus*
acclimated to warmer temperatures (Qu et al. 2018), while warming decreases cellular
requirements for iron in the nitrogen-fixing cyanobacterium *Trichodesmium* (Jiang et
al. 2018). In nitrogen-limited cultures of the marine diatom *Thalassiosira pseudonana*,
long-term thermal adaptation acted most strongly on systems other than those involved
in nitrate uptake and utilization (O'Donnell et al., 2018). Thus, it is possible that our
thermal response results with *E. huxleyii* might have differed under nutrient-limited



growth conditions.
***One-day vs two-day thermal variation***

As temperature fluctuations in the surface ocean increase along with climate

change, phytoplankton will be influenced by the frequencies and intensities of these
thermal excursions. We found that the responses of *E. huxleyi* to one-day versus two-
day temperature variations were different at both low and high temperature. For
instance, under low temperature the transition from the warm phase to the cool phase
during the thermal variation could be treated as a low temperature stress leading to a
lag phase in growth. The growth rate of the one-day variation treatment at the cool
phase was lower than that of the two-day variation, suggesting that physiological
acclimation is not rapid enough to accommodate to the shorter variation treatment,
while the two day variation allows enough time for growth to recover. However, at the
warm phase (21 °C) there was no difference in growth rates between the one-day and
two-day variations compared with the constant 21-degree treatment. These results
imply that there was a shorter lag phase after transfer at the optimal temperature point
(21 °C at the warm phase) than during low temperature stress (16 °C at the cool phase).

There was no significant difference in photosynthetic rates between the one-day

and two-day variation during the warm phase (21 °C), but both were higher than during
the cool phase, indicating the photosynthetic rate was correlated to the thermal
variation cycle. However, for the calcification rate there was no significant difference
between one-day and two-day variations during either the cool or warm phases. These
results suggested that photosynthesis was more responsive to temperature variations



than calcification, and so ultimately determined the growth rate in both cool and warm
phases. Feng et al. (2017) reported a similar relationship between growth and
photosynthetic rates of a Southern Hemisphere *E. huxleyi* cultured at different
temperatures.

Temperature variation frequencies also strongly influenced elemental

composition. In low temperature experiments, the cellular contents of PON, POP and
POC in the one-day variation treatment were all higher than under two-day variations.
A notable exception to this trend was the cellular PIC content, which was not
significantly different between one-day and two-day variation treatments. The PIC
content was positively correlated to calcification and relatively stable, indicating that
coccolith production and storage of *E. huxleyi* was relatively independent of the
frequency of thermal variation.

Unlike the photosynthetic rate, the cellular elemental content of one-day and two-

day variations were significantly different, but were not changed during temperature
variation when transitioning from the warm phase to the cool phase or vice versa.
The temperature dependent photosynthetic enzyme activity likely determined the
similar photosynthetic rate of one-day and two-day variation treatments at both cool
and warm phase in our short-term experiment, but the divergent responses of cellular
stoichiometry in one-day and two-day thermal variations indicated different
mechanisms of rapid acclimation to different thermal fluctuation frequencies. Our
results imply that the responses of *E. huxleyi* to one-day and two-day thermal
variations have different patterns, but both reach stable states during extended periods



of temperature fluctuation. Due to decreasing POC content, the PIC/POC ratio
increased in the two-day variation compared with the one-day variation, suggesting
that more rapid thermal fluctuations might lead to a decrease in calcite ballasting of
sinking organic carbon.

Under the high temperature scenario, thermal variation forces the microalgae to

intermittently deal with a lethal high temperature during the warm phase (28 °C), with
potentially irreversible damage to the cells.    In the "summer" experiments, the mean
growth rate of the two-day variation was much lower than that of the one-day variation.
This mainly resulted from the negative growth rate of two-day variation cultures
during the warm phase (28 °C), whereas the growth rate of the one-day variation
was >0.20 d$^{-1}$. This result demonstrates that high frequency temperature variations
(one-day) can partly mitigate growth inhibition by high temperatures in *E. huxleyi,* and
so allow tolerance to extreme thermal events relative to longer exposures. This
observation agrees with previous studies of other marine organisms such as corals
(Oliver & Palumbi, 2011; Safaie et al., 2018). In the case of our experiments, the lag
phase and metabolic inertia would help to maintain the microalgae during short
exposures (one-day) to high temperature when transitioning from the cool phase (23
°C) to the warm phase (28 °C).

Likewise, the particulate organic element contents (PON, POP and POC) of *E.*

*huxleyi* were more stable in one-day than in two-day temperature variation treatments.
The relatively steady status of cellular particulate organic matter content in the high
frequency temperature variation treatment may conserve energy, compared to the



energy-intensive redistribution of major cellular components under lower frequency
temperature variations. This differential energetic cost may help to explain the
differences in growth rates between the two treatments.    Adaptation to high
temperature may also require higher investment in repair machinery, such as heat shock
proteins, leading to an increased demand for nitrogen and other nutrients, thus
increasing cellular POC, PON and POP contents (O'Donnell et al., 2018).
**Prediction and modelling of *E. huxleyi* responses to thermal variation**

Mathematical curves based on population growth rates from laboratory studies

have been used to predict future population abundance, persistence or fitness in a
changing world (Bernhardt et al., 2018; Deutsch et al., 2008; Jiang et al. 2017). We
applied a modified version of the Eppley thermal performance curve to predict the
influence of thermal variation on the growth rate of *E. huxleyi* (Fig. 8). *E. huxleyi*
growth rates were predicted to be much lower at warmer temperatures under variable
conditions compared to constant conditions, but there were no significant differences
at cooler temperatures. Thus, the effect of thermal variation on population growth at
the upper thermal limit was predicted to be stronger than that in the lower portion of
the thermal range (Bernhardt et al., 2018; Sunday et al., 2012). This phenomenon has
been widely observed in ectothermic taxa (Dell et al., 2011), but this model for the
effect of thermal variation on population growth rate may lack the ability to predict
species responses at the extreme edges of their ranges (Bernhardt et al., 2018).

Our results showed that the measured effects of a variable thermal regime on *E.*

*huxleyi* growth rate fitted well with model-predicted values at a relatively low





temperature (mean=18.5 ºC), but differed considerably at high temperature (mean=25.5
ºC). This was especially evident under the two-day variation conditions at a mean of
25.5 ºC, where the growth rate was sharply lower than predicted from the constant
TPCs-based model. This result suggests that transient heat waves may erode thermal
tolerances of *E. huxleyi* populations already growing near their upper thermal limits,
and that the frequency and duration of such extreme events is critically important in
determining the magnitude of this stress. This sensitivity to increased thermal
variability may reduce the fitness of *E. huxleyi* in the future subtropical and tropical
oceans.
Although thermal variation at high temperature negatively impacted the growth
rate of *E. huxleyi* in our experiment, our relatively short-term study didn't address the
potential for *E. huxleyi* to evolve under selection by frequent extreme heat events.
Evolutionary change in the thermal optimum and the maximum growth temperature in
response to ocean warming may reduce heat-induced mortality, and mitigate some
ecological impacts of global warming (O'Donnell et al., 2018, Thomas et al., 2012). For
example, Schlüter et al. (2014) found that after one year of experimental adaptation to
warming (26.3°C), the marine coccolithophore *E. huxleyi* evolved a higher growth rate
when assayed at the upper thermal tolerance limit. Similar results were reported for the
marine diatom *Thalassiosira pseudonana* in recent studies (O'Donnell et al., 2018;
Schaum et al., 2018). Schaum et al. (2018) also found that the evolution of thermal
tolerance in marine diatoms can be particularly rapid in fluctuating environments.
Furthermore, populations originating from more variable environments are generally





more plastic (Schaum & Collins, 2014; Schaum et al., 2013). Long-term evolutionary
experiments with *E. huxleyi* will be necessary to determine how the thermal
performance curve of this important marine calcifier may diverge under selection by
different frequencies and durations of extreme thermal variation events.

Understanding the combination of ocean warming and magnified thermal

variability may be a prerequisite to accurately predicting the effects of climate change
on the growth and physiology of the key marine calcifier *E. huxleyi.* This information
will help to inform biogeochemical models of the marine and global carbon cycles, and
ecological models of phytoplankton distributions and primary productivity. How
changing thermal variation frequencies will affect marine phytoplankton remains a
relatively under-explored topic, but one that is likely to become increasingly important
in the future changing ocean.



**Acknowledgements**
Support was provided by U.S. National Science Foundation Biological Oceanography
grants OCE1538525 and OCE1638804 to F-XF and DAH. XW was supported by a
grant from the China Scholarship Council.



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



**Figure legends:**

**Fig. 1** Thermal performance curves (TPCs) showing cell-specific growth rates ($d^{-1}$) of *Emiliania huxleyi* CCMP371 across a temperature range from 8.5 to 28.6 ℃. Symbols represent means and error bars are the standard deviations of three replicates at each temperature, but in many cases the errors bars are smaller than the symbols.

**Fig. 2** Changes in *Emiliania huxleyi* TPC/PON ratios, POC/PON ratios, PIC/POC ratios and Cha/POC ratios across a temperatures range from 8.5 to 28.6 ℃. Dashed lines represent the average ratios for the entire temperature range. Bars represent means and error bars are the standard deviations of three replicates at each temperature. Symbols * represent the significant difference ($p < 0.05$) between average ratios and the ratio at each temperature.

**Fig. 3** *Emiliania huxleyi* growth rate responses to constant temperatures, and during the warm and cool phases of the two thermal variation frequencies (one-day and two-day), under low (**A**) and high (**B**) mean temperatures. The thick black line in the boxplots represent median values for each experimental treatment; whiskers on boxplots indicate 1.5 × interquartile range. Listed p-values with their respective brackets are the statistical significance between two treatments.

**Fig. 4** Responses of *Emiliania huxleyi* PIC/POC ratios to constant temperatures, and during the warm and cool phases of two thermal variation frequencies (one-day and two-day), under low (**A**) and high (**B**) mean temperatures. LT: Low temperature; HT: High temperature. The thick black line in the boxplots represent median values for each experimental treatment; whiskers on boxplots indicate 1.5 × interquartile range. Listed




p-values with their respective brackets denote the statistical significance between two
treatments.
**Fig. 5** Responses of *Emiliania huxleyi* total carbon fixation (photosynthesis +
calcification), photosynthetic and calcification rates to constant temperatures, and
during the warm and cool phases of two thermal variation frequencies (one-day and
two-day), under low (**A**, **C**, **E**) and high (**B**, **D**, **F**) mean temperatures. LT: Low
temperature; HT: High temperature. The thick black line in the boxplots represent
median values for each experimental treatment; whiskers on boxplots indicate 1.5 ×
interquartile range. Listed p-values with their respective brackets denote the statistical
significance between two treatments.
**Fig. 6** Responses of *Emiliania huxleyi* calcification to photosynthesis ratios (cal/photo)
to constant temperatures, and during the warm and cool phases of two thermal variation
frequencies (1 day and 2 day), under low (**A**) and high (**B**) mean temperatures. LT: Low
temperature; HT: High temperature. The thick black line in the boxplots represent
median values for each experimental treatment; whiskers on boxplots indicate 1.5 ×
interquartile range. Listed p-values with their respective brackets denote the statistical
significance between two treatments.
**Fig. 7** Responses of *Emiliania huxleyi* elemental ratios in two thermal variation
frequency treatments (1 day and 2 day) compared to constant temperatures, for:
TPC/PON (**A**, cool phase and **B**, warm phase), PON/POP (**C**, cool phase and **D**, warm
phase) and Chl *a*/POC ratios (**E**, cool phase and **F**, warm phase). LT: Low temperature;
HT: High temperature. The thick black line in the boxplots represent median values for





each experimental treatment; whiskers on boxplots indicate 1.5 × interquartile range.
Listed p-values with their respective brackets denote the statistical significance between
two treatments.
**Fig. 8** Thermal performance curves (TPCs) based on specific growth rates ($d^{-1}$) of
*Emiliania huxleyi*, including our experimentally determined constant temperature TPC
(black symbols and solid line) and an Eppley model-predicted fluctuating temperature
TPC (dashed line). Measured growth rates from the two low and high temperature
experiments are shown for constant thermal conditions (red symbols), one-day (green
symbols) and two-day (blue symbols) variation treatments.





Table 1 The effect of temperature variation under low and high temperature on total Carbon (pg/cell), cellular POC (pg/cell), cellular PIC (pg/cell), cellular PON (pg/cell), cellular POP (pg/cell) and cellular Chl *a* (pg/cell) of *Emiliania huxleyi*.

| Treatment | | Total Carbon | Cellular PON | Cellular POP | Cellular POC | Cellular PIC | Cellar Chl *a* |
|---|---|---|---|---|---|---|---|
| **low temperature** | 18.5 °C | 11.5±0.4 | 1.8±0.2 | 0.17±0.00 | 8.0±0.6 | 3.5±0.3 | 0.14±0.00 |
| | One-day cool point (16) | 13.0±0.5 | 2.2±0.3 | 0.18±0.00 | 8.9±0.3 | 4.1±0.3 | 0.15±0.01 |
| | One-day warm point (21) | 12.0±0.7 | 2.1±0.3 | 0.19±0.00 | 9.3±0.9 | 2.7±0.9 | 0.19±0.00 |
| | Two-day cool point (16) | 10.1±0.7 | 1.3±0.2 | 0.16±0.01 | 6.0±0.9 | 4.0±0.3 | 0.12±0.01 |
| | Two-day warm point (21) | 10.4±0.5 | 1.5±0.2 | 0.17±0.01 | 6.6±0.5 | 3.8±0.3 | 0.15±0.01 |
| **high temperature** | 25.5 °C | 15.0±0.7 | 2.0±0.1 | 0.21±0.01 | 9.5±0.3 | 5.5±0.7 | 0.18±0.02 |
| | One-day cool point (23) | 16.1±1.4 | 3.0±0.2 | 0.21±0.00 | 12.9±1.5 | 3.2±0.2 | 0.15±0.01 |
| | One-day warm point (28) | 19.1±0.8 | 4.4±0.3 | 0.24±0.01 | 17.0±0.6 | 2.1±0.2 | 0.20±0.02 |
| | Two-day cool point (23) | 12.4±1.0 | 1.9±0.2 | 0.19±0.01 | 7.5±1.0 | 4.8±0.3 | 0.13±0.01 |
| | Two-day warm point (28) | 19.4±2.0 | 3.9±0.8 | 0.25±0.03 | 18.3±3.7 | 2.1±0.9 | 0.25±0.02 |







**Fig. 1**

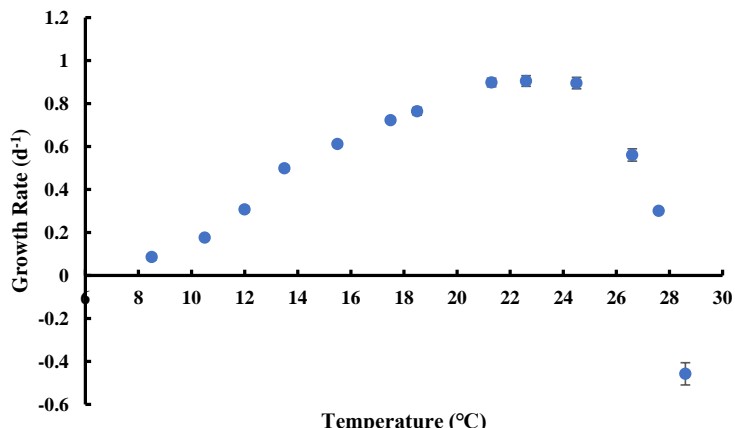




**Fig. 2**

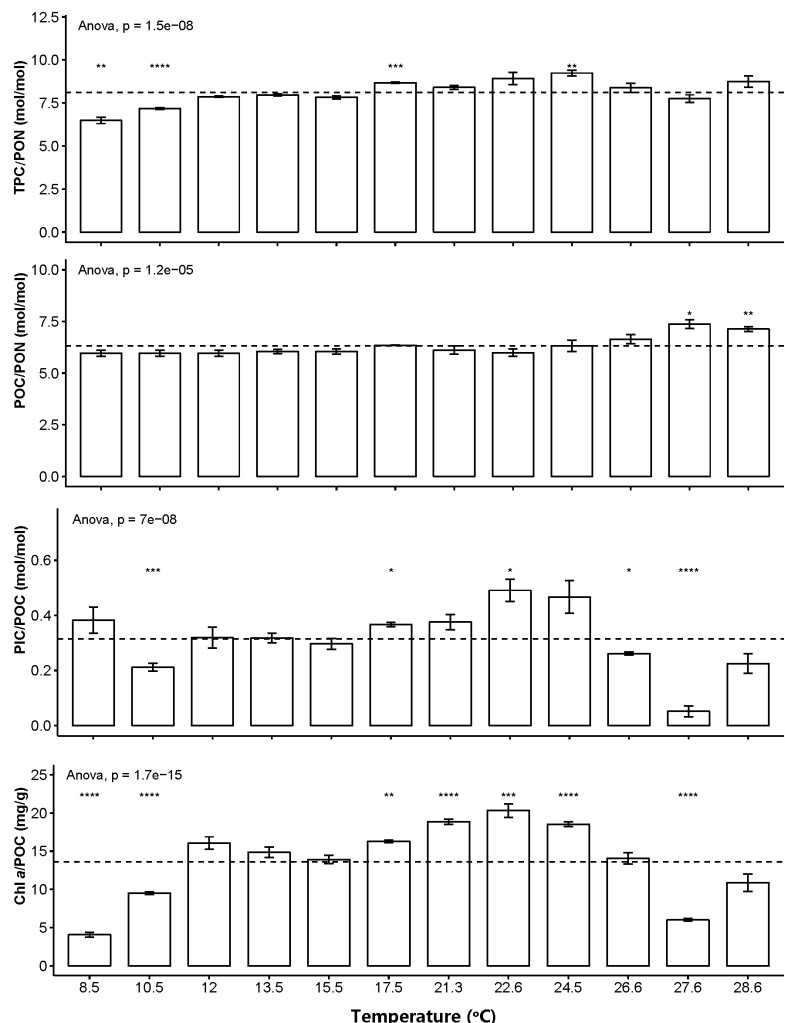







**Fig. 3**

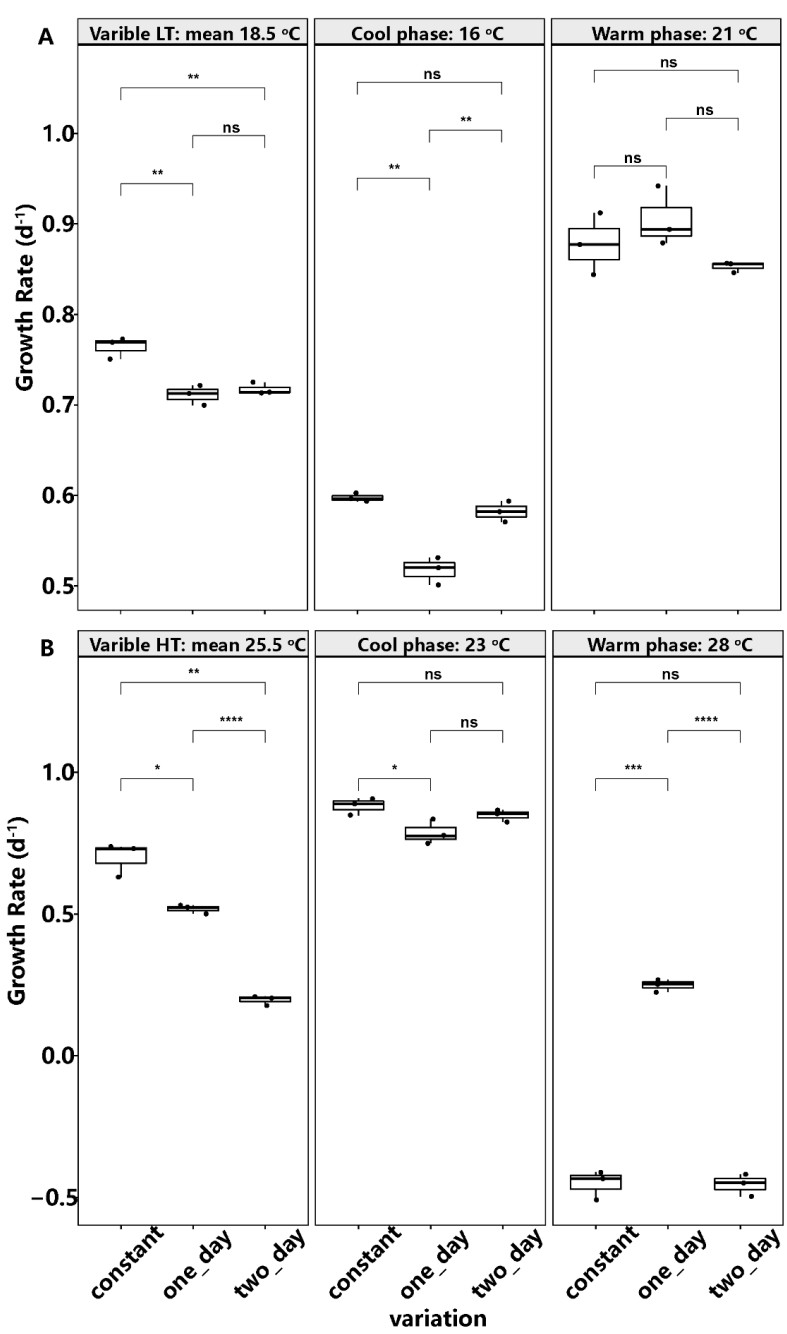






**Fig. 4**

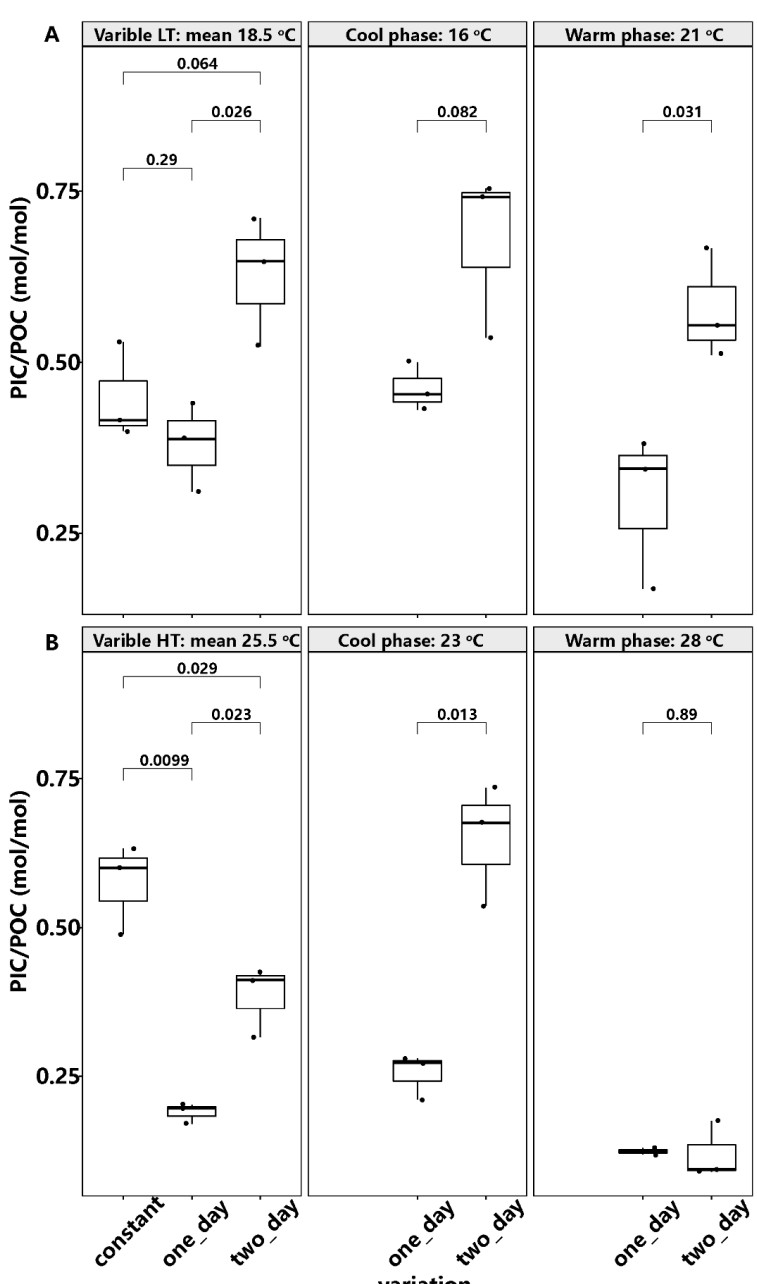





**Fig. 5**

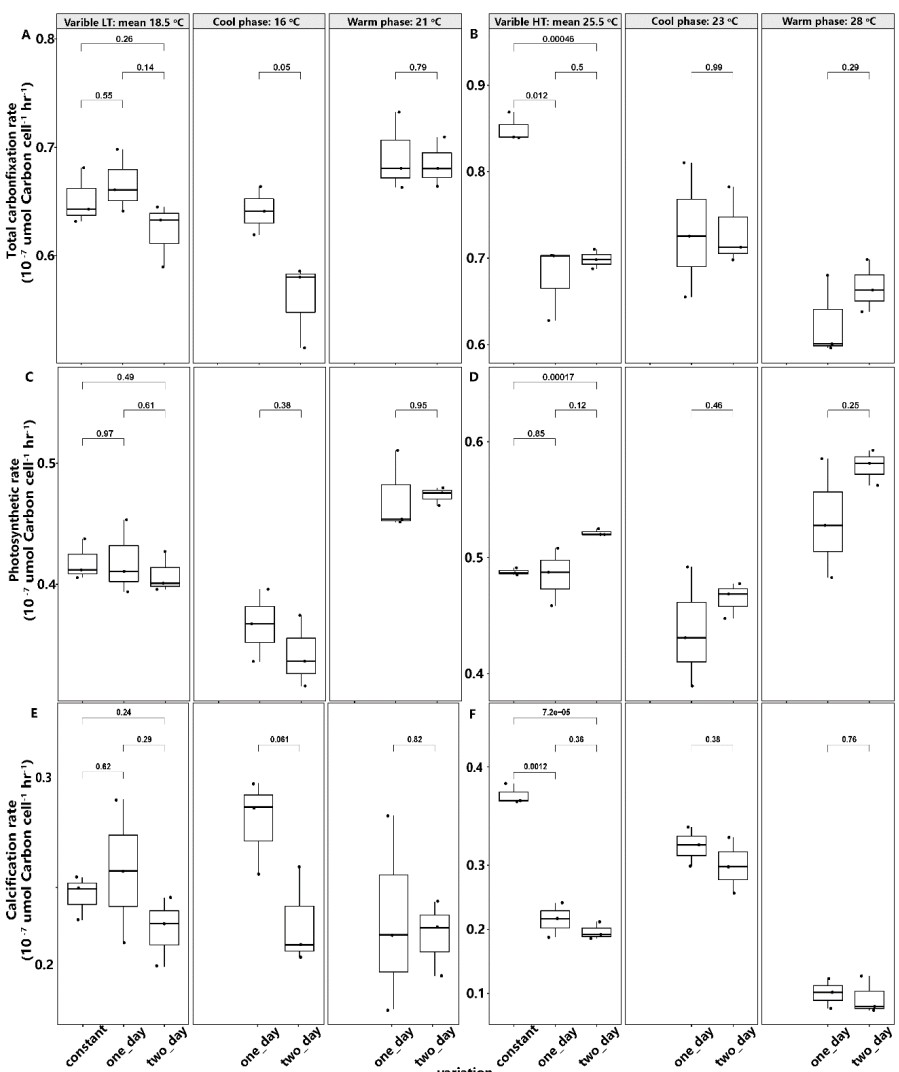





**Fig. 6**

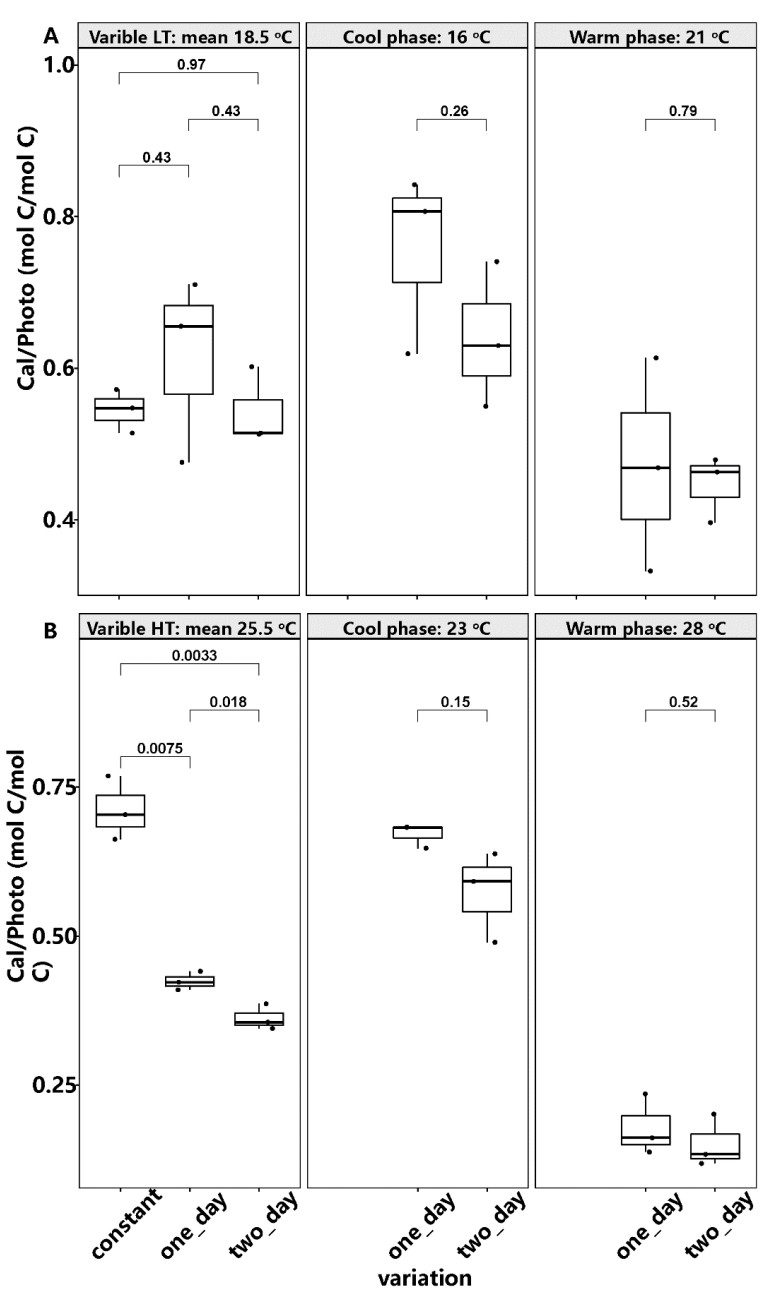







**Fig. 7**

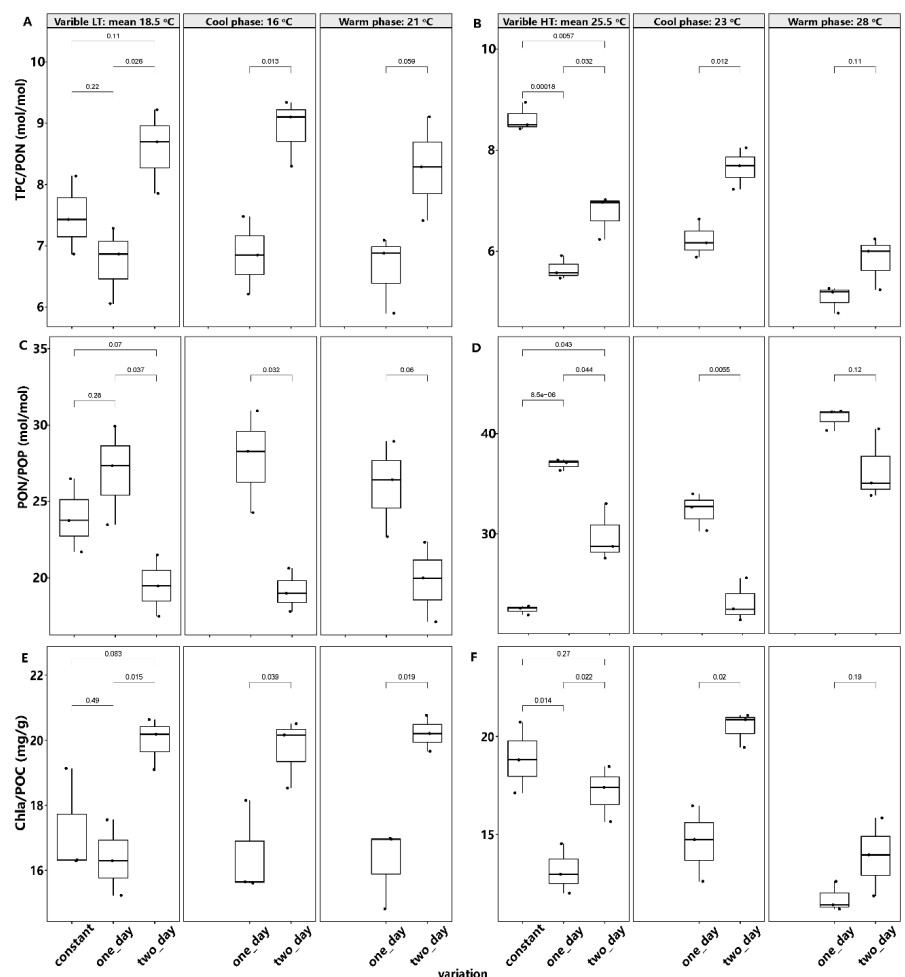







**Fig. 8**

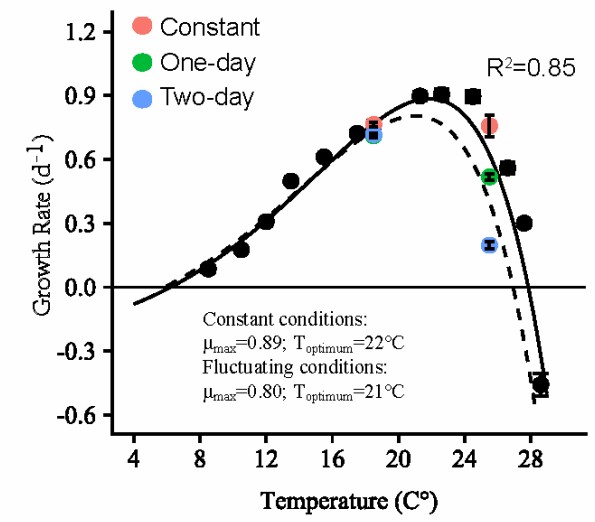
