# Peer review of "How will the key marine calcifier *Emiliania huxleyi* respond to a warmer and more"

_Biogeosciences, 2019_

## Referee Comment (RC1) · Anonymous Referee #1 · 1 Jul 2019

Review on: 'How will the key marine calcifier Emiliania huxleyi respond to a warmer and more thermally variable ocean?' by Wang et al.

The experiments are well designed and I have only a couple of smaller questions (see specific comments). The manuscript is well written. Overall, I found the discussion not extremely inspiring because I thought it missed a conceptual framework that helps to arrange the numerous datasets. Nevertheless, some of the key conclusions are interesting and the data is valuable. I therefore only have 'minor comments'

One major issue, however, is that the authors should deposit their data in a publicly accessible data repository and provide the link within the paper. This is important.

[Figure]

Specific comments:

Line 132: How was light measured and kept identical between treatments? Measuring light in such blocks is challenging and there may be large differences between replicates and treatments. Please provide a detailed description. Line 135: Was the dilution medium also Aquil? Please clarify. Line 139: It is unclear to me from this description how negative growth was measured. Wasn't it just the reduction in cell numbers or in your case red fluorescence? Please explain this better. Line 146: Please indicate how long it took for the temperature block to reach the new temperature after switching the water bath temperature. Is there a significant time lag? I wonder if this could partially explain the lower response in the one day cycle, as the time lag may have promoted a weaker response. Line 154: Weren't the nutrients already in the dilution medium? Or did you adjust to 100 and 10 $\mu$mol/L? This is confusing. Please clarify. Line 167: It remains unclear if you always measured both fluorescence and cell number or if this varied between treatments? Please clarify and ideally give the reader an idea how similar the growth rates were when determined with these two measurements. Line 180: Please provide percentage of the HCl acid. Was it 37%? In this case fuming overnight is fairly extreme and may perhaps breakdown POC? Line 185: Not 100% sure but I assume Fu et al., 2007 did not invent this protocol. Please provide original papers here and also for POC,PON above. Line 188: See previous comment. Line 217: The description of the applied statistical tests needs a better description. Perhaps briefly go through the consecutive steps. Just for completeness. Only mentioning which tests were done may raise some eye brows. Line 227: What is the rationale behind showing the TPC/PON ratio? What meaning does it have and why is it important? I would intuitively say that this dataset could be removed from the results but I am of course interested what the authors think. Line 266: This may indicate a time lag until the high temperature was established so that the warm period was shorter than indicated by assuming an instant change in temperature. Please provide a retention time for how long it lasted until the new temperature was reached within the bottles. Line 267: This comment basically addresses all quota measurements and ratios. When you look at e.g.

PIC/POC and do this for a one day period in the cycled experiments. To what extent is the response you measure and report here 'diluted' by the PIC/POC that manifested during the previous temperature that prevailed before? Is there a carry-over to the next day that needs to be accounted for? Line 380: The abbreviation TPCs is not ideal because it can be confused with total particulate carbon. I would suggest to use no abbreviation here. Line 406: A particularly comprehensive assessment was done by Zhang et al., 2014 from the Reusch group. This should definitely be considered here. Line 409: The Zhang et al., paper seems an overlooked but important paper here. Line 417: Schlueter et al., 2014 (also Reusch group) have shown that Ehux can quickly adapt to warming. Should be mentioned here, perhaps. Line 476: I don't understand how this trend can suggest these things. Isn't the damage of biochemical mechanisms simply your interpretation of what may have happened. Should be rephrased. Line 607: 'ectothermic' refers to animals or also plants/microbes? Please specify. Fig. 2 shows that the plasticity in PIC/POC is much larger than in the other ratios in this figure. I find this very interesting. Maybe it would be worth discussing this issue. Fig 6B: y-axis incomplete.

I hope my suggestions help the authors to improve their manuscript.

––––––––––––––––––––––––––––

---

## Referee Comment (RC2) · Anonymous Referee #2 · 27 Aug 2019

Temperature is an important driver regulating phytoplankton physiology. Previous laboratory and field investigations suggest that the trend of global warming may strongly affect future phytoplankton communities and the consequent marine biogeochemistry. Most previous studies of warming effects on phytoplankton were mainly conducted under relatively constant temperature regimes. However, under future climate change scenario, in addition to warming (i.e. increasing mean temperature), the magnitude of temperature fluctuation will also be changed. The response pattern of marine phytoplankton to thermal variations/fluctuations is still largely unknown. The present study investigated the physiological response of a well-studied marine coccolithophore species Emiliania huxleyi to not only a broad range of temperature regime, but also

two different frequencies (one-day and two-day) of thermal variation. The examined physiological parameters include growth, photosynthetic and calcification rates, and elemental compositions. The results suggest that higher thermal variation frequency (one-day) was less inhibitory on E. huxleyi physiological processes than two-day variations especially under high temperature, indicating that the frequency of temperature fluctuation may be of importance in regulating the impacts of extreme high temperature events on key phytoplankton groups. The conclusions are valuable and help to predict the relevant marine biogeochemistry under a more realistic condition of a complex and changing marine environment. In general, the manuscript is well written and organized; the results are also well explored and discussed. I would suggest the manuscript to be accepted with minor revisions. My detailed comments and suggestions are listed below.

Line 140: How often were these cultures diluted? Does this mean that steady-state growth was not observed for 28.6°C treatment?

Lines 144-148: For the different fluctuation cycles (one-day and two-day), how was the temperature adjusted? Was temperature changed gradually during a one-day or two-day period or the cultures experienced abrupt temperature changes? Was there any lag phase for temperature changes? It would be better to provide the details of temperature fluctuation patterns in different treatments in order to better explain the observed different effects of fluctuation frequencies on Emiliania huxleyi physiology.

Lines 152-155: What was the nutrient condition in the culture medium used for dilution? What do you mean by "100 $\mu$mol L-1 nitrate and 10 $\mu$mol L-1 phosphate was added every two days"? Please clarify.

Line 170: Please delete "GFC"

Lines 174-176: "Total Particulate Carbon" and "Particulate Organic Carbon/Nitrogen" should all be lowercased.

[Figure]

Line 206: I found the abbreviation of "TPC" a bit confusing here, since it refers to "total particulate carbon" in the earlier text.

Line 209: misspelling of Emiliania huxleyi

Line 210: Please specify how the equation was modified.

Line 251: Please rephrase the text to "The growth rates during the cool phase of the one-day variation cycle were lower than those..."

Line 419: should be revised to " can be influenced...".

Line 596 - : In this section, it might be worth to also expand the discussion on how thermal variation would affect the competition advantage of coccolithophores over other phytoplankton functional groups (such as diatoms) in the community level.

Fig. 1. The growth rates presented in the figure were supposed to be measured during steady growth phase. However, according the context, the cultures were not able to survive at 28.6°C. I assume the negative growth rate was calculated based on the decreased in-vivo fluorescence values over the consecutive sampling days. I'd suggest using the value 0 instead of negative value for fitting at this data point.

---

## Author Response (AR1)

**Response to Editor**

Associate Editor Decision: Publish subject to minor revisions (review by editor) (07 Oct 2019) by Julia Uitz
Comments to the Author:
Dear authors,

My apologies for the long delay in getting back to you regarding the status of your paper.

I believe overall you have satisfactorily addressed the comments and questions raised by both Reviewers. I will be pleased to accept your paper for publication in Biogeosciences, subject to minor changes being made in response to my comments provided below.

Please consider these minor additional comments and include appropriate changes in the revised version of your manuscript. Then upload the final version of your ms onto the BG interface.

Sincerely,
Julia

Response to editor:
Dear Dr. Julia Uitz, Thank you very much for your letter and the additional comments about our paper.
In response to the reviewers' and your comments we have made numerous revisions to our manuscript. We have provided the detailed responses to the comments of the reviewers and you, including a point-by-point reply to the comments.
Additionally, we have modified the first author's affiliations and added Yahui Gao as a co-corresponding author for his contribution on this paper.
We submit here the revised manuscript (with a red color font to mark-up the changes made in the manuscript).
We would like to express our great appreciation again to you for your comments on our paper. Looking forward to hearing from you.
* * *
Additional comments
Please consider the comments below, revise the text and correct all typos before you upload the revised version of your manuscript. I've listed several typos below but note that the list is not exhaustive.

The line numbering of the text in red color in the revised text is often wrong, which doesn't help to identify how/if you respond appropriately to the Reviewers' comments.

Unfortunately we neglected to check the line numbers in the response letter against the final revised manuscript. We certainly apologize for this oversight, as we understand it is important for you and the reviewers to be able to locate our revisions quickly and easily. We've also corrected and proofread the manuscript for additional typos, thank you for those that you pointed out.

Responses to RC#1
RC1-Line 210 : Please specify how the equation was modified.
Your response that the wording has been clarified so as to more accurately describe the model in l. 252-255 is confusing and likely refers to the wrong line numbers. The section "Model for population growth…" is in l. 230-239 and includes only minor changes compared to the submitted version. Please make sure you properly accommodate this specific comment by the Reviewer in your revised ms.

Response:   We have now expanded this methods text with a much more in-depth description of how the Bernhardt et al. (2018) model works, and why it gives superior estimates under variable thermal regimes compared to older linear models (**lines 230-243**). In addition, we have provided several references for readers who would like to learn the derivation and application of the model for themselves.

RC1-Line 596- : In this section, it might be worth to also expand the discussion…in the community level.
In your response, you mention that changes were made into the text in l. 656-661 but I was not able find any change inhere. Please make sure this important suggestion is accounted for in the revised version of the ms.

Response: We apologize for submitting a revised version with wrong line numbers, we understand this makes it more difficult to review our changes and it shouldn't have happened. Now, we have expanded the existing (but mis-numbered) discussion about effects of thermal variability on potential competition with other phytoplankton taxa such as the tropical diatoms and cyanobacteria found in the same region as *E. huxleyi,* and point out that this may affect both their relative fitness and community structure. Please check **Line 678-684** in the present revised version.

RC1-Fig 1 : Negative growth rates.
I understand your response. Yet I believe the biological meaning of these negative growth rates should be discussed as other readers may have the same impression as the Reviewers.

Response: We expanded the text about the potential biological meaning and usefulness of the negative growth rates as suggested, and explained our rationale for presenting them (rather than as zero growth rates) in the same fashion that we explained to the reviewer in our response letter.   At the end of this paragraph, there is text discussing ecological implications of these negative growth rates for the marine coccolithophore *E. huxleyi* as the Sargasso Sea (where this strain was isolated) continues to warm. (**Lines 458-469**).

Responses to RC#2
Line 217: The description of the applied statistical tests needs a better description…
In response to this comment, I can see you've simply added the name of the R functions you used to perform the statistical tests. This does not accommodate the Reviewer's comment. I believe she/he is expecting you to indicate the steps in your calculations. By the way, in l. 246 ("two formulas including compare_means() and stat_compare_mean()"),you actually refer to "functions" from the R package, not mathematical formula so please correct the text accordingly.

Response: Thank you for this good suggestion. We have revised the text accordingly, including listing the specific stepwise procedures used in our statistical analyses, as well as the R functions (not formulas, we corrected this) used to calculate them. (**Lines 248-260**)

Typos
l. 139 : please clarify « fluorescent lights were rearranged »
Response: Following your suggestions, we have revised in the manuscript to clarify this. (**Lines 133- 139**)
l. 168 : remove "the" in « very two days the for constant »
Response: We revised the manuscript as suggested. (**Line 167**)
l. 173 : consider adding « at » into the following sentence « always maintained >100 μmol L-1 and >10 μmol L-1 »
Response: We revised the manuscript as suggested. (**Line 172**)
l. 199 : correct « rations »
Response: We revised this mis-spelling. (**Line 196**)
l. 212 : remove « s » at the end of "techniques" in « using a 14C incubation techniques »

or correct sentence appropriately

Response: We followed this suggestion and have revised the manuscript. (**Line 209-210**)

l. 362 : Please define abbreviation or write in full "Cal/Photo ratios"

Response: We followed this suggestion and have revised the manuscript. (**Line 373**)

l. 459: Remove "s" at the end of "rises"

Response: We revised the manuscript as suggested. (**Line 474**)

l. 597: A word must be missing here "of a Southern Hemisphere cultured at"

Response: We added the missing words '*E. huxleyi* strain'. (**Line 607**)

l. 676: Please write in full "didn't address »

We followed this suggestion and have revised the manuscript. (**Line 686**)

Acknowledgement section: Please consider acknowledging the Reviewers for their constructive comments and suggestions.

Response: We agree this is a good idea, and have added acknowledgement of the Reviewers for their constructive comments and suggestions. (**Lines 728-729**)

**Response to Anonymous Referee #1**
Review on: 'How will the key marine calcifier *Emiliania huxleyi* respond to a warmer and more thermally variable ocean?' by Wang et al. The experiments are well designed and I have only a couple of smaller questions (see specific comments). The manuscript is well written. Overall, I found the discussion not extremely inspiring because I thought it missed a conceptual framework that helps to arrange the numerous datasets. Nevertheless, some of the key conclusions are interesting and the data is valuable. I therefore only have 'minor comments' One major issue, however, is that the authors should deposit their data in a publicly accessible data repository and provide the link within the paper. This is important.

Response: The authors would like to thank the anonymous Reviewers for their constructive comments and suggestions to improve the quality of the paper. Those comments are all valuable and very helpful for revising and improving our paper. We have studied comments carefully and have made correction which we hope meet with approval. Revised portion are marked in red in the paper. The main corrections in the paper and the responds to the reviewer's comments are as flowing:

Response to Reviewer #1 (highlights):

Thank you very much for your helpful comments. Our data from this paper have been submitted to the Biological and Chemical Oceanography Data Management Office (BCO-DMO, bco-dmo.org), as is required by the conditions of our major funding agency (US NSF). The data are currently in the queue to be uploaded, but the data management office is running behind and we have been told that it will be several months more before the data can be quality checked, vetted and formatted, and posted to be made publicly available. When this is finished, the data will be available at our project webpage: www.bco-dmo.org/project/668547. We can provide this link with the paper if the editor agrees, but it will still take some time before the data from this paper are live.

Response to Reviewer #1 (Specific comments):

Line 132: How was light measured and kept identical between treatments? Measuring light in such blocks is challenging and there may be large differences between replicates and treatments. Please provide a detailed description.

Response: We agree that getting the lighting uniform for every replicate within a thermal block is essential but can be difficult, and we went to considerable effort to carefully measure and adjust light levels in each position in the block to be as close to identical as possible. We followed your suggestion, and now provide a detailed section in the Methods on how we measured and adjusted the light intensity in the thermal-blocks. (**Line 133-139**)

Line 135: Was the dilution medium also Aquil? Please clarify.

Response: Yes, the Aquil medium was used as the dilution medium, and have now we clarified this in the manuscript. (**Line 141**)

Line 139: It is unclear to me from this description how negative growth was measured. Wasn't it just the reduction in cell numbers or in your case red fluorescence? Please explain this better.

Response: Yes, the negative growth rate was calculated from the decrease of cell numbers at 28.6 °C during cultivation. In our preliminary experiments, we repeated this process several times to rigorously verify that cultures were unable to grow at this temperature. We have revised and expanded the description of how negative growth rates were measured in our manuscript. (**Line 146-150**)

Line 146: Please indicate how long it took for the temperature block to reach the new temperature after switching the water bath temperature. Is there a significant time lag? I wonder if this could partially explain the lower response in the one day cycle, as the time lag may have promoted a weaker response.

Response: This is an important point. It took the block about half an hour to re-adjust to the transformed temperature for each growth phase, which shouldn't represent a significant time lag relative to the 24-48 h thermal cycles. The reason for the lower response to the one-day cycle is likely the acclimation characteristics of the coccolithophorid. We have revised and clarified the description of the thermal cycles and their re-adjustment times during transitions in the manuscript. (**Line 158-163**)

Line 154: Weren't the nutrients already in the dilution medium? Or did you adjust to 100 and 10 _mol/L? This is confusing. Please clarify.

Response: Thanks for pointing this out, we agree this text was unclear and confusing. We did adjust the N and P midway through the 4 day cycle (2 day variation treatment) by adding concentrated Aquil stocks at these concentrations to make sure nutrients remained replete throughout the 4 day cycle. We have revised this text in the manuscript to better describe this. (**Line 168-172**)

Line 167: It remains unclear if you always measured both fluorescence and cell number or if this varied between treatments? Please clarify and ideally give the reader an idea how similar the growth rates were when determined with these two measurements.

Response: Following your suggestions, we have revised in the manuscript to clarify this. (**Line 184-186**) Under constant conditions such as in the thermal block and the constant controls of the variation treatment, the cell numbers and the *in vivo* fluorescence are strongly correlated and relatively invariant (as verified by microscopic counts). So, we used the *in vivo* fluorescence to calculate the growth rate. However, the cellular *in vivo* fluorescence (cellular Chl *a* content) changed during temperature fluctuation, so for these treatments we applied cell counts only to calculate the growth rate.

Line 180: Please provide percentage of the HCl acid. Was it 37%? In this case fuming overnight is fairly extreme and may perhaps breakdown POC?

Response: We revised in the manuscript to provide this information (Line **200**)

In our experiment, we used the ~37% saturated HCl for fuming overnight to thoroughly remove the inorganic carbon. We are not aware of any published evidence that ~12h of HCl fuming can degrade organic carbon, but we can consider this possibility if the reviewer knows of any. From our results shown in Fig. 2C, the PIC/POC ratio was extremely low (~0.05), meaning that the POC content was nearly as high as the TPC (PIC+POC) content. This result suggests that the cellular POC is very likely not degraded by our saturated HCl fuming method.

Line 185: Not 100% sure but I assume Fu et al., 2007 did not invent this protocol. Please provide original papers here and also for POC, PON above.

Response: We gave our own references for these methods because in our lab over the years we have made minor modifications to these classic protocols, and this allows readers to look up the exact procedures we used if desired. However, in response to this suggestion we have revised the manuscript by adding the original citations as well for all of these methods (Line **202, 204-206**)

Line 188: See previous comment.

Response: As noted above, we have now added citations to the original protocols preceding the citations of our slightly modified versions of the techniques. (Line **209-210**)

Line 217: The description of the applied statistical tests needs a better description. Perhaps briefly go through the consecutive steps. Just for completeness. Only mentioning which tests were done may raise some eye brows.

Response: We followed this suggestion and have revised in the manuscript to include a better and more in-depth description of the statistical methods. (**Line 248-260**)

Line 227: What is the rationale behind showing the TPC/PON ratio? What meaning does it have and why is it important? I would intuitively say that this dataset could be removed from the results but I am of course interested what the authors think.

Response: We understand that many coccolithophore studies don't present the

TPC/PON ratio, but we feel it is worth presenting as it encompasses all of the C fixed (into both POC and PIC) relative to all of the cellular N quota. We also of course present the more traditional POC:PON and PIC:POC ratios as well.

Line 266: This may indicate a time lag until the high temperature was established so that the warm period was shorter than indicated by assuming an instant change in temperature. Please provide a retention time for how long it lasted until the new temperature was reached within the bottles.

Response: The time for the thermal block to re-equilibrate the experimental bottles after temperatures were switched was only half an hour, which we suggest is too short to significantly affect the overall growth rates in either the one day or two day thermal variation treatments. We have revised in the manuscript with new text to point this out. (Line **309-311**)

Line 267: This comment basically addresses all quota measurements and ratios. When you look at e.g. PIC/POC and do this for a one day period in the cycled experiments. To what extent is the response you measure and report here 'diluted' by the PIC/POC that manifested during the previous temperature that prevailed before? Is there a carry-over to the next day that needs to be accounted for?

Response: We have considered this phenomenon during our experiment, so during dilutions we replaced a large proportion of the culture with fresh medium (up to 80-90%) to avoid significant carry-over from the old growth phase. The ideal condition of course would be to switch from cool phase to warm phase and then cycle without dilution. However, this is impossible as dilution with fresh medium is necessary to avoid nutrient limitation setting in and confounding our results. In addition, volume removed for sampling needs to be replaced with fresh medium in our relatively small volume experimental thermal block setup.

Line 380: The abbreviation TPCs is not ideal because it can be confused with total particulate carbon. I would suggest to use no abbreviation here.

Response: We followed this suggestion and have revised the manuscript to avoid using the abbreviation TPC here, as indeed we had already used to stand for total particulate carbon. Instead, here we now write out the words 'temperature performance curve' (**Line 424-432, 457, 675, 1050-1052**).

Line 406: A particularly comprehensive assessment was done by Zhang et al., 2014 from the Reusch group. This should definitely be considered here.

Response: We followed this suggestion and have revised in the manuscript to include the Zhang et al. reference. (**Line: 452**)

Line 409: The Zhang et al., paper seems an overlooked but important paper here.

Response: As noted above, we have revised in the manuscript to include consideration of the Zhang et al. 2014 study. (Line: **453-456**)

Line 417: Schlueter et al., 2014 (also Reusch group) have shown that Ehux can quickly adapt to warming. Should be mentioned here, perhaps.

Response: We talk about rapid adaptation to warming in the following section, and we have already cited the Schlueter et al. 2014 study in this context. (**Line: 691**)

Line 476: I don't understand how this trend can suggest these things. Isn't the damage of biochemical mechanisms simply your interpretation of what may have happened. Should be rephrased.

Response: We agree that we should be more specific and support our suggestion with evidence from the literature. Accordingly, we have revised in the manuscript to point out that energetic and material investments in cellular repair machinery such as heat shock proteins are needed to deal with stressfully high temperatures, and supported this statement with a new reference (O'Donnell et al. 2018). (Line: **532-536**)

Line 607: 'ectothermic' refers to animals or also plants/microbes? Please specify.

Response: We have now stated that we are specifically referring to ectothermic animals at this point in the manuscript. We agree that even though plants and microbes can't control their body temperature either, the term ectotherm is usually reserved for animals. (Line: **667**)

Fig. 2 shows that the plasticity in PIC/POC is much larger than in the other ratios in this figure. I find this very interesting. Maybe it would be worth discussing this issue.

Response: This is an insightful comment, and so we have added new text to the Discussion to point out the large plasticity in PIC:POC ratios with temperature changes, and to discuss this observation in terms of a prior study by Krumhardt et al. (2017), as well as pointing out potential implications for ballasting of sinking particles. (**Line 499-505**)

Fig 6B: y-axis incomplete.

Response: The Y-axis scale in Fig 6 has now been extended to 1.0 in order to encompass all of the data points, thanks for pointing this out.

I hope my suggestions help the authors to improve their manuscript.

We do appreciate the constructive comments of the reviewer, and they have indeed improved the paper.

**Response to Anonymous Referee #2**
Temperature is an important driver regulating phytoplankton physiology. Previous laboratory and field investigations suggest that the trend of global warming may strongly affect future phytoplankton communities and the consequent marine biogeochemistry. Most previous studies of warming effects on phytoplankton were mainly conducted under relatively constant temperature regimes. However, under future climate change scenario, in addition to warming (i.e. increasing mean temperature), the magnitude of temperature fluctuation will also be changed. The response pattern of marine phytoplankton to thermal variations/fluctuations is still largely unknown. The present study investigated the physiological response of a well-studied marine coccolithophore species *Emiliania huxleyi* to not only a broad range of temperature regime, but also two different frequencies (one-day and two-day) of thermal variation. The examined physiological parameters include growth, photosynthetic and calcification rates, and elemental compositions. The results suggest that higher thermal variation frequency (one-day) was less inhibitory on *E. huxleyi* physiological processes than two-day variations especially under high temperature, indicating that the frequency of temperature fluctuation may be of importance in regulating the impacts of extreme high temperature events on key phytoplankton groups. The conclusions are valuable and help to predict the relevant marine biogeochemistry under a more realistic condition of a complex and changing marine environment. In general, the manuscript is well written and organized; the results are also well explored and discussed. I would suggest the manuscript to be accepted with minor revisions. My detailed comments and suggestions are listed below.

Response: We appreciate the reviewer's thoughtful comments and enthusiasm for our study, and have described our revisions and responses to their helpful comments below.

Line 140: How often were these cultures diluted? Does this mean that steady-state growth was not observed for 28.6_C treatment?

Response: The cultures were diluted every two days the for constant and one-day variation treatments, and every four days for two-day variation treatments. (Methods, **Line 166-168**). The reviewer is correct, since a negative growth rate was calculated from the decrease of cell numbers at 28.6 °C during cultivation, the coccolithophore was unable to survive at this temperature, and growth was not at steady state- this treatment could not be diluted due to the declining biomass, and thus represents a batch culture rather than a semi-continuous one. To be certain that 28.6 °C exceeded the upper thermal limit, we repeated the experiment at this temperature several times. We have discussed this with new text on **Line 146- 150**

Lines 144-148: For the different fluctuation cycles (one-day and two-day), how was the temperature adjusted? Was temperature changed gradually during a one-day or two-day period or the cultures experienced abrupt temperature changes? Was there any lag phase for temperature changes? It would be better to provide the details of temperature fluctuation patterns in different treatments in order to better explain the observed different effects of fluctuation frequencies on *Emiliania huxleyi* physiology.

Response: The temperature setting of the thermal block setup was switched over fully (not gradually) at each transition between fluctuation cycles, but took about ½ hour to equilibrate to the new temperature after being changed, thus allowing some time for the cells to acclimate to the temperature shift. We did not observe any significant growth rate lag following the thermal shifts, just a rapid transition to a new growth rate. We have provided a detailed description in the manuscript. (**Line 158-163**)

Lines 152-155: What was the nutrient condition in the culture medium used for dilution? What do you mean by "100 _mol L-1 nitrate and 10 _mol L-1 phosphate was added every two days"? Please clarify.

Response: We adjusted the N and P midway through the 4 day cycle (2 day variation treatment) by adding concentrated Aquil stocks at these final concentrations to make sure nutrients were replete.    We have revised this text in the manuscript to better describe this. (**Line 168-172**)

Line 170: Please delete "GFC"

Response: We have revised in the manuscript as suggested. (**Line 189**)

Lines 174-176: "Total Particulate Carbon" and "Particulate Organic Carbon/Nitrogen" should all be lowercased.

Response: We have made this change. (**Line 193-195**)

Line 206: I found the abbreviation of "TPC" a bit confusing here, since it refers to "total particulate carbon" in the earlier text.

Response: The abbreviation TPC for 'thermal performance curve' has been removed here, since the reviewer is correct, it was used earlier in the paper for 'total particulate carbon'. Thermal performance curve is now written out. (**Line 424-432, 457, 675, 1050-1052**).

Line 209: misspelling of *Emiliania huxleyi*
Response: We revised this mis-spelling. (**Line 225, 230**)

**Line 210: Please specify how the equation was modified.**

Response: Our approach used for predicting thermal response curves under variable thermal conditions (as opposed to the constant temperatures used in the classic Eppley study) was first published by Bernhardt et al. (2018). It is a non-linear averaging model that incorporates the principle of Jensen's inequality, and so is based on Eppley's equation but with these modifications to deal with fluctuating temperatures. It has been applied in published thermal variation studies by Qu et al (2019) and Kling et al. (in press), both cited here. The full derivation of this thermal variation model is too lengthy to give here, but can be obtained by interested readers from the Bernhardt paper. We changed the original confusing wording to more accurately describe this model on **Line 230-243.**

Line 251: Please rephrase the text to "The growth rates during the cool phase of the one-day variation cycle were lower than those…"
Response: We followed this suggestion and have revised the manuscript. (**Line 293-294**)

Line 419: should be revised to " can be influenced…".
Response: We revised the manuscript as suggested. (**Line 471**)

Line 596 - : In this section, it might be worth to also expand the discussion on how thermal variation would affect the competition advantage of coccolithophores over other phytoplankton functional groups (such as diatoms) in the community level.

Response: Thank you for this good suggestion. We have revised the discussion text accordingly. (**Line 678-684**)

Fig. 1. The growth rates presented in the figure were supposed to be measured during steady growth phase. However, according the context, the cultures were not able to survive at 28.6_C. I assume the negative growth rate was calculated based on the decreased in-vivo fluorescence values over the consecutive sampling days. I'd suggest using the value 0

instead of negative value for fitting at this data point.

Response: As noted above, the negative growth rate was calculated from the decrease of cell numbers at 28.6 °C during cultivation during a batch culture, an experiment which we repeated several times to robustly verify this result. The magnitude of the negative growth rate here is an expression of the degree of stress the culture experienced at this temperature, and may be useful to some readers for comparison with the other positive growth rate values in the variation experiments. We appreciate the comment, but with the editor's permission would like to keep the negative value here.

[revised manuscript text omitted]